# FUSIONVIT: HIERARCHICAL 3D OBJECT DETECTION VIA LIDAR-CAMERA VISION TRANSFORMER FUSION

## ABSTRACT

For 3D object detection, both camera and lidar have been demonstrated to be useful sensory devices for providing complementary information about the same scenery with data representations in different modalities, e.g., 2D RGB image vs 3D point cloud. An effective representation learning and fusion of such multi-modal sensor data is necessary and critical for better 3D object detection performance. To solve the problem, in this paper, we will introduce a novel vision transformer-based 3D object detection model, namely FusionViT. Different from the existing 3D object detection approaches, FusionViT is a pure-ViT based framework, which adopts a hierarchical architecture by extending the transformer model to embed both images and point clouds for effective representation learning. Such multi-modal data embedding representations will be further fused together via a fusion vision transformer model prior to feeding the learned features to the object detection head for both detection and localization of the 3D objects in the input scenery. To demonstrate the effectiveness of FusionViT, extensive experiments have been done on real-world traffic object detection benchmark datasets KITTI and Waymo Open. Notably, our FusionViT model can achieve the state-of-the-art performance and outperforms not only the existing baseline methods that merely rely on camera images or lidar point clouds, but also the latest multi-modal image-point cloud deep fusion approaches.

## 1 INTRODUCTION

Developing an efficient 3D object detection model is core for autonomous driving. Lidar and cameras are two commonly used sensors in this task (Li et al., 2016; Ku et al., 2017). Meanwhile, the sensor data obtained by Lidar and camera are in totally different modalities, i.e., sparse point clouds vs RGB images, which is very difficult to handle and fuse together within one model for the object detection task (Arnold et al., 2019). Pure image based (Girshick, 2015; Carion et al., 2020; Liu et al., 2021; Wang et al., 2022) and pure point cloud based (Zhou & Tuzel, 2017; Lang et al., 2018; Yin et al., 2020; Zhou et al., 2023c;a; 2022) strategies have exploited their raw data structure characteristics in the object detection task, whose performance is still far below the safety requirements for their deployment in our modern autonomous driving vehicles (Committee, 2021).

Modern autonomous driving vehicles pose higher and strict standards for object detection models in the perspective of both detection accuracy and model robustness. A single modality often cannot fully describe the complex correlation between data. However, image and point cloud can in some way complement each other: image information can focus on textual conditions, while point cloud data will not be affected by the light condition and also carries the scenery depth (Arnold et al., 2019). As a result, there are scenarios where it is hard to detect with just one type of sensor. Employing multi-modal data to describe the same scene from different perspectives helps extract comprehensive features to make the object detector more robust. Viewed from such a perspective, how to effectively combine such multi-modal sensory data while preserving the essential features of each modality becomes the key challenge.

Recently, transformers (Vaswani et al., 2017) have been widely deployed in Natural Language Processing (NLP) tasks (Devlin et al., 2018; Brown et al., 2020; Yang et al., 2019) with state-of-the-art performance. After ViT (Dosovitskiy et al., 2020), transformers started to be utilized on various vision-related research tasks, such as image classification (Touvron et al., 2020; Heo et al., 2021;

Fang et al., 2021), 2D (Carion et al., 2020; Liu et al., 2021) and 3D object detection (Misra et al., 2021; Wang et al., 2021b; Zhu et al., 2022; Zhou et al., 2023a; 2022; Bai et al., 2022).

Given that ViT was the initial CV application of the transformer model in the field of vision, it could have great potential for broad application scenarios. However, its uses are primarily restricted to (Touvron et al., 2020; Fang et al., 2021; Heo et al., 2021). In the field of 3d object detection, most state-of-the-art frameworks are not from ViT itself but its derivatives, like DETR-based methods (Wang et al., 2021b; Misra et al., 2021) which contain both transformer encoder and decoder, or Zhou et al. (2023a; 2022) that modified a lot from the original ViT prototype.

*Could it also exist a pure-ViT based framework that enhances the performance of 3D object detection?* That original model, indeed, may not be in the best settings for the 3D object detection task. The computational complexity of ViT increases quadratically with respect to image size, making it a severe issue to be used on a lidar point cloud, which contains thousands of points in a simple frame. In this paper, we propose a hierarchical vision transformer-based lidar-camera fusion strategy for object detection called **FusionViT** to relieve that issue so that it could reach promising performance, especially in traffic scenery. Based on the multi-modal camera image and lidar point cloud inputs, FusionViT includes a **CameraViT** and a **LidarViT** for learning the embedding representations of input data in each modality, respectively. Their learned representation will be further fused together for hierarchical and deeper representation learning via a novel **MixViT** component.

By partitioning images into mini-patch for representation learning, our proposed CameraViT extends the ViT (Dosovitskiy et al., 2020) model to object detection task, showing competitive 2D object detection performance. In addition, by partitions point clouds into mini-cubic, we introduce a novel LidarViT as a volumetric-based 3D object detector. Using a transformer encoder as the end-to-end backbone, it discretizes the point cloud into equally spaced 3D grids. It has great advantages in terms of helping the feature extraction network to be computationally more effective and reducing memory needs while keeping the original 3D shape information as much as possible. The most important contribution attributes to our proposed FusionViT model, bridging as consistent as possible between the fusion model and pre-fusion models, Benefiting from the transformer encoder model, our proposed FusionViT suits greatly under traffic scenes, where input frames are continuously changing spatially and timely.

Our model is not like previous methods (Chitta et al., 2022; Ku et al., 2017; Chen et al., 2016) that combine 2D features extracted from image or projection (such as Bird-eye-view, Range view, or Depth image), which introduces information bottleneck from these 3D to 2D projection operations. By directly doing end-to-end fusion, our model is also not necessary to explicitly set pixel-to-point-cloud alignment like Deepfusion (Li et al., 2022b) did, while outperforming it, showing the strong robustness of our strategy. In short, our main contributions are listed as follows:

- We are the first study to investigate the possible pure-ViT based 3D object detection framework to the best of our knowledge.

- We proposed a hierarchical pure-ViT based lidar-camera fusion framework for it called FusionViT, which exploited the inherent features between image and point cloud.

- We performed a vit-based detector on both 2D image detection and 3D point cloud detection respectively, leading to great performance under each part.

- Extensive experiments are conducted on the Waymo Open Dataset and KITTI benchmarks. FusionViT achieves competitive results compared with existing well-designed approaches, showing a promising future of pure-ViT based frameworks in 3D object detection tasks.

## 2 RELATED WORK

### 2.1 2D OBJECT DETECTION

2D object detection locate objects of interest in 2D images and classify them into pre-defined categories. The advances in deep learning have revolutionized the field of 2D object detection. The R-CNN (Girshick et al., 2013) and its extensions (Girshick, 2015; Ren et al., 2015; He et al., 2017) used a Region Proposal Network to generate candidate object proposals and a Convolutional Neural Network (CNN) for object classification. Besides these two-stage detectors, single-stage detectors

like SSD (Liu et al., 2016), YOLO serious (Redmon et al., 2015; Redmon & Farhadi, 2016; 2018; Bochkovskiy et al., 2020; Li et al., 2022a; Wang et al., 2022) simultaneously classifies anchor boxes and regresses the bounding boxes at the same time, which are usually more efficient in terms of inference time while comparably get slightly lower accuracy. The revolution from the transformer (Vaswani et al., 2017; Dosovitskiy et al., 2020) brings more powerful 2D detectors (Carion et al., 2020; Liu et al., 2021), which further increases the competitive edge.

## 2.2 3D Object Detection in Point Clouds

Represented as an unordered set, point cloud provides richer information about the environment, enabling more accurate and robust object detection. Early methods (Qi et al., 2016; 2017b) directly apply Neural Networks on the raw points. (Karlinsky et al., 2020; Shi et al., 2018) also learn features using PointNet-like layers. Projection-based methods first project point cloud into 2D representation, such as range images (Meyer et al., 2019; Sun et al., 2021), then use Neural Networks to predict 3D bounding boxes. Volumetric-based methods convert lidar points to voxels (Yan et al., 2018; Zhou & Tuzel, 2017) or pillars (Lang et al., 2018; Yang et al., 2021). These three tracks are primarily exploited by modern cutting-edge approaches (Zhou et al., 2023a;c; 2022; Shi et al., 2020). Some of them (Sun et al., 2022; Fan et al., 2021; 2022) also take advantage of sparse mechanics (Yan et al., 2018)to reduce onerous computational demands while retaining excellent detection accuracy. Others (Misra et al., 2021; Zhang et al., 2022; Wang et al., 2021b; Liu et al., 2022a) focus on making full use of transformers (Vaswani et al., 2017) by bridging its natural gap for 3D point cloud.

## 2.3 Lidar-Camera Fusion

The key task of Lidar-Camera fusion should be the feature alignment between Point Cloud and Camera Image. Along this idea, (Vora et al., 2019; Sindagi et al., 2019; Wang et al., 2021a) match each point cloud of camera images, extracting features from the camera images to decorate the raw point clouds. State-of-the-art approaches (Prakash et al., 2021; Bai et al., 2022; Zeng et al., 2022; Liu et al., 2022b; Zhou et al., 2023b) are primarily based on LiDAR-based 3D object detectors and strive to incorporate image information into various stages of a LiDAR detection pipeline since LiDAR-based detection methods perform significantly better than camera-based methods. Combining the two modalities necessarily increases computing cost and inference time lag due to the complexity of LiDAR-based and camera-based detection systems. As a result, the problem of effectively fusing information from several modes still exists.

## 3 Proposed Methods

### 3.1 Notations and Terminologies

In the sequel of this paper, we will use the upper or lower case letters (e.g., $X$ or $x$) to represent scalars, lower case bold letters (e.g., $\mathbf{x}$) to denote column vectors, and bold-face upper case letters (e.g., $\mathbf{X}$) to denote matrices, and upper case calligraphic letters (e.g., $\mathcal{X}$) to denote sets or high-order tensors. We use $\mathbf{X}^\top$ and $\mathbf{x}^\top$ to represent the transpose of matrix $\mathbf{X}$ and vector $\mathbf{x}$. The concatenation of vectors $\mathbf{x}$ and $\mathbf{y}$ of the same dimension is represented as $\mathbf{x} \sqcup \mathbf{y}$.

Figure 1 shows the overall architecture of the proposed FusionViT. FusionViT accepts multi-modal inputs, which include both RGB images and point clouds. The input images are defined as $\mathcal{I} \in \mathbb{R}^{H_I \times W_I \times 3}$, where $H_I$ and $W_I$ denote the image height and width dimensions, respectively. Meanwhile, the input point cloud is represented as a set of 3D points $\mathcal{P} \in \mathbb{R}^{H_P \times W_P \times D_P}$ in the $H_P \cdot W_P \cdot D_P$ 3D space, where each point in it is a vector of its $(x, y, z)$ coordinate.

As shown in Figure 1, our FusionViT model has a hierarchical architecture. Given camera images and lidar point cloud as inputs, some necessary data prepossessing is needed to produce the 2D image embedding and 3D point cloud embedding, respectively. Based on the partitioned image and point cloud batches, we introduce a CameraViT model to learn the image embedding and a LidarViT to learn the 3D point cloud embedding, respectively. Their learned representation will be combined together and fed as the inputs of the mix component for representation fusion and learning. Taking its output, a MixViT model will further fuse the learned embedding. An object Detection head is

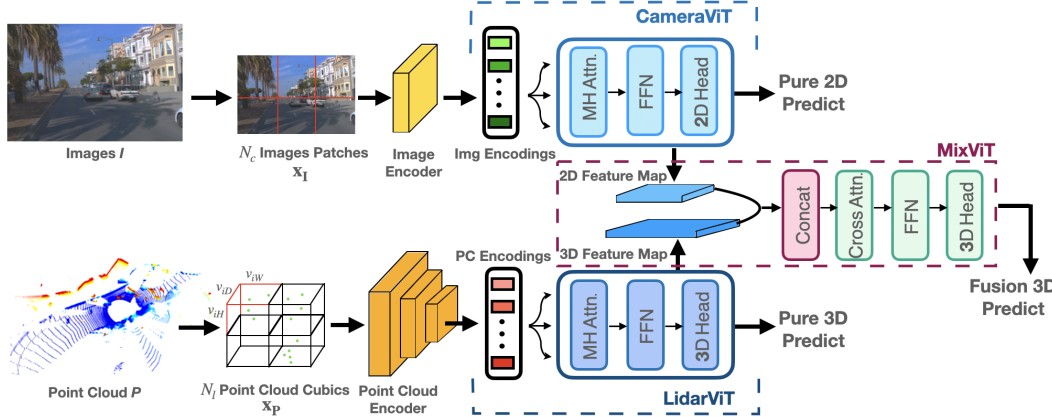

Figure 1: The Whole Framework of FusionViT

finally added to detect the objects existing in the input scenery data. In this section, we will introduce those aforementioned functional components in detail for readers.

## 3.2 CAMERAVIT BASED IMAGE EMBEDDING

Camera as a sensory device can capture the scenery information with RGB images. Object detection and localization from images have been studied for many years (Girshick et al., 2013; Liu et al., 2016; Redmon et al., 2015). In recent years, transformer (Vaswani et al., 2017) based models have been demonstrated to outperform conventional CNN models (Lecun et al., 1998) in solving many vision problems (Dosovitskiy et al., 2020). In this paper, we propose CameraViT, which extends the ViT model to the traffic scenery object detection problem setting, by partitioning images into mini-patch for representation learning.

Formally, within the CameraViT model, by setting each patch to have the side length $v_{cH}$ and $v_{cW}$ respectively, it partitions each image $\mathbf{I} \in \mathbb{R}^{H_I \times W_I \times 3}$ image into $N_c$ 2D patches $\mathbf{X_I} \in \mathbb{R}^{N_c \times (v_{cH} \cdot v_{cW} \cdot 3)}$, where $N_c = \frac{H_I}{v_{cH}} \frac{W_I}{v_{cW}}$ denotes the patch number.

Such partitioned image patches will be flattened into one-dimensional $v_{cH} \times v_{cW} \times 3$ features, then fed into a Multi-Layer Perceptron (MLP) to reach in total $N_c$ encoded features. For each encoded features $i$, we have

$$\mathbf{x}_c^i = \mathrm{MLP}\left(\mathbf{x}_I^i\right). \tag{1}$$

The MLP's parameters are shared by all patches so that they are encoded in the same way. Each encoded feature $\mathbf{x}_c^i$ has the same vector size $D_c$, which is also the output size of the MLP.

Similar to BERT's [class] token (Devlin et al., 2018), we prepend a learnable embedding $\mathbf{E_c}$ to the sequence of the encoded patch features $\left(i.e.\mathbf{z}_0^0 = \mathbf{x}_{class}\right)$. Position embeddings $\mathbf{E}_{posc}$ are added to the patch embeddings to retain positional information, i.e. $\mathbf{Z}_0 = \left[\mathbf{x}_{class}; \mathbf{x}_c^1\mathbf{E_c}; \mathbf{x}_c^2\mathbf{E_c}; \cdots; \mathbf{x}_c^N\mathbf{E_c}\right] + \mathbf{E}_{posc} \in \mathbb{R}^{(N_c+1) \times D_c}$. The resulting sequence of embedding vectors serves as input to the Camera Transformer Encoder. The encoder consists of alternating layers of Multiheaded Self-Attention (MSA) (Vaswani et al., 2017) and MLP blocks. Layernorm (LN) is applied before every block, and residual connections after every block:

$$\begin{cases} \mathbf{Z}_l' = \mathrm{MSA}\left(\mathrm{LN}\left(\mathbf{Z}_{\ell-1}\right)\right) + \mathbf{Z}_{\ell-1} \\ \mathbf{Z}_l = \mathrm{MLP}\left(\mathrm{LN}\left(\mathbf{Z}_\ell'\right)\right) + \mathbf{Z}_\ell' \end{cases}, \forall l = 1, 2, ..., L_c, \tag{2}$$

where $L_c$ is the layer number of the CameraViT model. The final image represented features $\mathbf{H}_c$ are generated from the Layernorm of the output of the Transformer encoder $\mathbf{Z}_{L_c}^0$:

$$\mathbf{H}_c = \mathrm{LN}\left(\mathbf{Z}_{L_c}^0\right), \tag{3}$$

where $\mathbf{H}_c \in \mathbb{R}^{h_c \times N_c}$, assuming $h_c$ is the hidden size of the camera transformer encoder. The learned features $\mathbf{H}_c$ could complete pure 2D Object Detection tasks by adding an object detection head following it. It also has the potential to concatenate with other learned features from different sensors to perform multi-modal prediction.

### 3.3 LidarViT based Point Cloud Embedding

Transformers (Vaswani et al., 2017) is well suited for operating on 3D points since they are naturally permutation invariant and some transformers-based 3D Object Detectors (Misra et al., 2021; Zhang et al., 2022; Wang et al., 2021b; Zhu et al., 2022; Liu et al., 2022a; Zhou et al., 2023a; 2022) are proposed to solve the 3D vision problem, gaining promising performance.

On the other hand, ViT was the first work to apply the transformer model in the vision field. That model, however, may not be in the best settings for the 3D object detection task. The computational complexity of ViT increases quadratically with respect to image size, making it a severe issue to be used on a lidar point cloud, which contains thousands of points in a simple frame. *Could it also exist a pure-ViT based structure that enhances the performance of 3D object detection?* Inspired by the Voxel Feature Encoding layer in (Zhou & Tuzel, 2017), we designed a voxel-based LidarViT to resolve the huge memory burden, making it more suitable for point cloud data structure.

LidarViT process the point cloud input from Lidar. Inspired by 2D image processing operations, we here divide the raw point cloud in the whole 3D space into little cubic, each with side length $P_l$. Different from images, Point Cloud is a sparse representation, which means most of these cubic is empty. We therefore first remove these empty cubic to cut down the computation burden. The number of non-empty Point Cloud cubic $N_l$ will also be the input length of the LidarViT model.

We conduct Random Sampling for those cubic having a large number of point cloud. Typically a high-definition LiDAR point cloud is composed of about 100k points. Directly processing all the points not only imposes increased memory/efficiency burdens on the computing platform, but also highly variable point density throughout the space might bias the detection. To this end, we randomly sample a fixed number, T, of points from those cubic containing more than T points. This sampling strategy has two purposes, (1) computational savings; and (2) decrease the imbalance of points between the cubic which reduces the sampling bias, and adds more variation to training.

In each cubic, a flatten and a different MLP operation will be conducted. Raw point cloud is an unordered set, where each point is independent of others. Transformer encoders, on the other hand, are suited greatly to some kind of dependent data structure (like sentences, and images). Therefore, augmenting raw point cloud data could be one key idea to make full use of the Transformer structures. Inspired by the Voxel Feature Encoding layer in (Zhou & Tuzel, 2017), we use a $K$-layer hierarchical feature encoding process for each point cloud cubic.

Setting each cubic has the side length $v_{lH}$, $v_{lW}$ and $v_{lD}$ respectivel, it transforms the whole $\mathcal{P} \in \mathbb{R}^{H_P \times W_P \times D_P}$ point cloud space into $N_l$ 3D cubics $\mathbf{X_P} \in \mathbb{R}^{N_l \times (v_{lH} \cdot v_{lW} \cdot v_{lD})}$, where: $N_l = \frac{H_P}{v_{lH}} \frac{W_P}{v_{lW}} \frac{D_P}{v_{lD}}$ denotes the cubic number.

For $\mathbf{X_P}' = \left\{ \mathbf{x_P^i} = [x_i, y_i, z_i]^T \in \mathbb{R}^3 \right\}_{i=1...t}$ as a non-empty cubic containing $t \leq T$ LiDAR points, where $\mathbf{x_P^i}$ contains XYZ coordinates for the $i$-th point. We first compute the local mean as the centroid of all the points in $\mathbf{X_P}'$, denoted as $(c_x, c_y, c_z)$. Then we augment each point $\mathbf{x_P^i}$ with the relative offset w.r.t. the centroid and obtain the input feature set $\mathbf{X_{Pin}} = \left\{ \hat{\mathbf{x_P^i}} = [x_i, y_i, z_i, x_i - c_x, y_i - c_y, z_i - c_z]^T \in \mathbb{R}^6 \right\}_{i=1...t}$. Each cubic will then be flattened into one-dimensional $v_{lH} \times v_{lW} \times v_{lD}$ features, and fed into a Multi-Layer Perceptron (MLP). For each encoded features $i$, we have

$$\mathbf{x}_l^{i'} = \mathrm{MLP}(\hat{\mathbf{x_P^i}}). \tag{4}$$

Then, each $\mathbf{x_l^{i'}}$ will be transformed through a fully connected network (FCN) into a feature space $\mathbf{f}_i$. The FCN is composed of a linear layer, a batch normalization (BN) layer, and a Sigmoid Linear Units (SILU) layer. After obtaining point-wise feature representations, element-wise MaxPooling is used across all $\mathbf{f}_i$ to get the locally aggregated feature $\tilde{\mathbf{f}}_i$. Finally, each $\mathbf{f}_i$ with $\mathbf{f}_i$ are combined to form the point-wise concatenated feature as $\mathbf{x}_l^i$. In this way, we could output the encoded features $\mathbf{X}_l = \left\{ \mathbf{x}_l^i \right\}_{i...t}$:

$$\mathbf{f}_i = \mathrm{FCN}(\mathbf{x}_l^{i'}), \tilde{\mathbf{f}}_i = \mathrm{MAX}(\mathbf{f}_i), \mathbf{x}_l^i = \left[ \mathbf{f}_i^T, \tilde{\mathbf{f}}_i^T \right]^T. \tag{5}$$

The FCN and MLP parameters are shared by all non-empty cubic so that they are encoded in the same way. Each encoded feature $\mathbf{x}_l^i$ has the same vector size $D_l$.

We prepend another learnable embedding $\mathbf{E_l}$ to the sequence of the encoded patch features. Position embeddings $\mathbf{E}_{posl}$ are added to the patch embeddings to retain positional information, i.e. $\mathbf{z}_0 = \left[\mathbf{x}_{class}; \mathbf{x}_l^1\mathbf{E_l}; \mathbf{x}_l^2\mathbf{E_l}; \cdots ; \mathbf{x}_l^N\mathbf{E_l}\right] + \mathbf{E}_{posl} \in \mathbb{R}^{(N_l+1)\times D_l}$. The resulting sequence of embedding vectors serves as input to the Lidar Transformer Encoder. Replacing $L_c$ into the layer number of Lidar Transformer Encoder $L_l$, formula 2 is applied. The final learned point cloud represented features $\mathbf{H}_l$ are generated from the Layernorm of the output of the Transformer encoder $\mathbf{Z}_{L_l}^0$:

$$\mathbf{H}_l = \mathrm{LN}\left(\mathbf{Z}_{L_l}^0\right), \qquad (6)$$

where $\mathbf{H}_l \in \mathbb{R}^{h_l \times N_l}$, assuming $h_l$ is the hidden size of the lidar transformer encoder. Similar to $\mathbf{H}_c$, $\mathbf{H}_l$ could complete pure 3D Object Detection tasks by adding an object detection head following it, as well as performing multi-modal prediction by concatenating with other learned features possibly from different sensors.

## 3.4 MixViT: CameraViT and LidarViT Fusion

Another issue of keeping from directly using the naive ViT in the image-point cloud multi-modal detection task is the feature fusion compatibility from the image and point cloud branch. Lidar data provides accurate geometric information about the scene, while camera data provides rich texture and color information. The learned representation from CameraViT and LidarViT should be fused together smoothly. We, therefore, designed a MixViT as well as a hierarchical structure to make the sparse like point cloud data better compatible with the dense like image data. Being as consistent as possible between the fusion model and pre-fusion models, we purpose MixViT to cut down feature misalignment and model incompatibility, improving the accuracy and robustness.

MixViT concatenates the learned feature representation from image $\mathbf{H}_c = N_c + N_l$ and point cloud $\mathbf{H}_l$. Then use another MLP to reach $N_m$ numbers of encoded features $\mathbf{X_m}$:

$$\mathbf{X_m} = \mathrm{MLP}(\mathbf{H}_c \sqcup \mathbf{H}_l). \qquad (7)$$

Here we assume the camera transformer encoder and the lidar transformer encoder have the same hidden size (i.e. $h_c = h_l = h$). The MLP's parameters are shared by all patches so that they are encoded in the same way. Each encoded feature $\mathbf{x}_m^i$ has the same vector size $D_m$. We have tried other fusion strategies but find this *concat* operation performs the best.

Similar to before, a learnable embedding $\mathbf{E_m}$ to the sequence of encoded patch features is prepended, and Position embeddings $\mathbf{E}_{posm}$ are added: $\mathbf{Z}_0 = \left[\mathbf{x}_{class}; \mathbf{x}_m^1\mathbf{E_m}; \mathbf{x}_m^2\mathbf{E_m}; \cdots ; \mathbf{x}_m^N\mathbf{E_m}\right] + \mathbf{E}_{posm} \in \mathbb{R}^{(N_m+1)\times D_m}$. The resulting sequence of embedding vectors serves as input to the Mix Transformer Encoder. Replacing $L_c$ into the layer number of Mix Transformer Encoder $L_m$, formula 2 is applied. The final learned point cloud represented features $\mathbf{H}_m$ are generated from the Layernorm of the output of the Transformer encoder $\mathbf{z}_{L_m}^0$:

$$\mathbf{H}_m = \mathrm{LN}\left(\mathbf{Z}_{L_m}^0\right), \qquad (8)$$

where $\mathbf{H}_m \in \mathbb{R}^{h_m \times N_m}$, assuming $h_m$ is the hidden size of the lidar transformer encoder. By adding an object detection head on the first vector $\mathbf{h}_m^0$, 3D Object Detection tasks could be completed.

## 3.5 Object Detection Head and Loss Function

Once getting the transformer output, we add a bounding box head and a classification head on the backbone. Both heads are Multi-Level Perceptions. The number of the hidden layer could be fine-tuned. The bounding box head outputs $\hat{\mathbf{U}} \in \mathbb{R}^{N \times O}$ features, where $N$ is the maximum prediction number. Each feature could be represented as $\hat{\mathbf{u}}_\mathbf{i} = (cx, cy, cz, l, w, h, \theta)$ in 3D object detection, where $O$ is 7, $\hat{\mathbf{u}}_\mathbf{i}^\mathbf{c} = (cx, cy, cz)$ represent the center coordinates, $\hat{\mathbf{u}}_\mathbf{i}^\mathbf{s} = (l, w, h)$ are the length, width, height, and $\hat{\mathbf{u}}_\mathbf{i}^\mathbf{h} = \theta$ denotes the heading angle in radians of the bounding box. In 2D object detection, each feature could be represented as $\hat{\mathbf{u}}_\mathbf{i} = (cx, cy, w, h)$, where $O$ is 4. The classification head outputs $\hat{\mathbf{V}} \in \mathbb{R}^{N \times (C+1)}$ features, where C is the number of classes. 1 is added for the "no object" class. Each $\hat{\mathbf{v}}_\mathbf{i}$ is the predicted probability of the object belonging to the positive class. We set the ground truth bounding box features to be $\mathbf{U}$, and ground truth class features to be $\mathbf{V}$, where each $\mathbf{v}_\mathbf{i}$ is a one-hot encoder setting the class label to be 1 and others being 0.

In our object detection task, the **total loss** is decomposed into two individual losses: classification loss and regression loss:

$$\mathcal{L}_{Total} = \lambda_1 \mathcal{L}_{cls} + \lambda_2 \mathcal{L}_{reg}. \tag{9}$$

The **classification loss** measures the error in predicting the object class label. The focal loss(Lin et al., 2017) function is used here which is a modified version of the cross entropy loss:

$$\mathcal{L}_{cls} = -\sum_i [\mathbf{v_i}(1 - \mathbf{\hat{v}_i})^\gamma \log(\mathbf{\hat{v}_i}) + (1 - \mathbf{v_i})\mathbf{\hat{v}_i}^\gamma \log(1 - \mathbf{\hat{v}_i})], \tag{10}$$

where $\gamma$ is a modulating factor that controls the weight given to each example. The **regression loss** measures the error in predicting the bounding box location (including center, size, and heading) of the object. Since it is common to have sparse outliers in $\mathbf{\hat{u}_i}$ and $\mathbf{u_i}$, inspired from (Meyer, 2019), we regard them two Laplace distributions to help improve robustness to outliers. Then we use the Kullback-Leibler divergence between each two Laplace distributions to compute the location loss:

$$
\begin{aligned}
\mathcal{L}_{reg} &= \mathcal{L}_{center} + \mathcal{L}_{size} + \mathcal{L}_{heading} + \lambda_3 \mathcal{L}_{corner} \\
&= \sum_i \mathrm{KL}_{Laplace}(\mathbf{\hat{u}_i^c}, \mathbf{u_i^c}) + \sum_i \mathrm{KL}_{Laplace}(\mathbf{\hat{u}_i^s}, \mathbf{u_i^s}) \\
&\quad + \sum_i \mathrm{KL}_{Laplace}(\mathbf{\hat{u}_i^h}, \mathbf{u_i^h}) + \lambda_3 \mathcal{L}_{corner}.
\end{aligned}
\tag{11}
$$

Note that center, size, and heading have separate loss terms, which may result in not optimized learning for final 3D box accuracy. Inspired by (Qi et al., 2017a), we add the corner loss $\mathcal{L}_{corner}$ to jointly optimized for best 3D box estimation under 3D IoU metric. the corner loss is the sum of the distances between the eight corners of a predicted box and a ground truth box. Since corner positions are jointly determined by center, size, and heading, the corner loss can regularize the multi-task training for those parameters.

### 3.6 FUSION-VIT WITH PRE-TRAINING AND FINE-TUNING

The overall structure of the hierarchical ViTs' 3D object detection model consists of the proposed FusionViT framework. However, there are three ViTs built hierarchy in total, which may lead to some potential issues of large memory consumption or heavy use. To eliminate these concerns, we also pre-train CameraViT and LidarViT on the training set. Then fine-tune to smaller tasks (like using a smaller testing dataset) using the whole framework. Adding object detection head, we first pre-train CameraViT by letting it read camera images and directly output 2D object detection prediction, then we pre-train LidarViT by letting it read point cloud data and directly output 3D object detection prediction. After that, we run the whole framework using the pre-trained CameraViT and LidarViT, and read the task images and point cloud data.

## 4 EXPERIMENTS

We compared our model with several baselines. For the baseline choosing, we selected both classical and SOTA models, typically for Object Detection tasks in camera-only, lidar-only, and camera-lidar fusion input. We show our model's inherent advantages of its design and structure over several current genres. The experiment setup and implementation details can be found on A and B.

### 4.1 EXPERIMENT RESULT ON WAYMO OPEN DATASET

On Waymo Open Dataset (Sun et al., 2019), We conduct three groups of experiments: pure 2D Detection from Camera Images, pure 3D Detection from Lidar Point Cloud, and 3D Detection from Camera Images and Lidar Point Cloud. Their performance on Waymo Open Dataset is shown in Table 1, as listed in the first, second, and last block, respectively. We report the Vehicle and Pedestrian's AP and APH scores. APH is only available in 3D Object Detection.

In pure 2D Detection, CameraViT out-performance state-of-art Transformer-based methods DETR (Carion et al., 2020) and Swim-Transformer (Liu et al., 2021), which shows promising results. In

| Model | Veh. | | Ped. | |
|---|---|---|---|---|
| | AP | APH | AP | APH |
| DETR (2021) | 48.5 | \ | 46.3 | \ |
| Swim-Transformer (2022) | 48.8 | \ | 49.2 | \ |
| **CameraViT** (ours) | **49.3** | \ | **49.8** | \ |
| PointPillars (2019) | 50.9 | 50.4 | 50.3 | 51.3 |
| CenterPoint (2020) | 53.2 | 53.3 | 53.7 | 53.2 |
| CenterFormer (2022) | 54.8 | 54.3 | 54.0 | 54.3 |
| SST (2022) | 54.4 | 55.0 | 54.5 | 54.0 |
| **LidarViT** (ours) | **55.3** | **55.3** | **54.6** | **54.8** |
| DeepFusion (2022) | 57.3 | 57.4 | 55.2 | 56.8 |
| TransFusion (2022) | 58.1 | 58.2 | 58.3 | 57.1 |
| **FusionViT** (ours) | **59.5** | **58.4** | **58.5** | **58.9** |
| **FusionViT Pretraining** | **60.3** | **60.1** | **61.4** | **59.8** |

Table 1: Comparison of attained validation AP and APH (in %) on Waymo Open Dataset

pure 3D Detection, LidarViT performs better than classical PointPillars (Lang et al., 2018) and CenterPoint (Yin et al., 2020) methods. It's also better than other state-of-art voxel-based frameworks CenterFormer (Zhou et al., 2022) and SST (Fan et al., 2021). The promising results show great use of the Transformer encoder. In the 3D Detection from both Camera Image and Lidar Point Cloud, our FusionViT out-performance the state-of-the-art method DeepFusion (Li et al., 2022b) and TransFusion (Bai et al., 2022). Although all are fusion-based, FusionViT takes good use of the learned features from pure 2D and pure 3D learning. In addition, the pre-trained version reaches higher performance than its original with around 50% shorter time. This shows good robustness and flexibility of the FusionViT model.

## 4.2 EXPERIMENT RESULT ON KITTI

| Model | $\text{mAP}_{\text{BEV}}$ (IoU $= 0.7$) | | | $\text{mAP}_{\text{3D}}$ (IoU $= 0.7$) | | |
|---|---|---|---|---|---|---|
| | Easy | Med | Hard | Easy | Med | Hard |
| PV-RCNN (2020) | 86.2 | 84.8 | 78.7 | N/A | N/A | N/A |
| Part-$A^2$-free (2021) | 88.0 | 86.2 | 81.9 | 89.0 | 72.5 | 69.4 |
| OcTr (2023) | 89.5 | 82.4 | 77.3 | 87.3 | 75.5 | 75.4 |
| FastPillars (2023) | 88.2 | 83.2 | 81.1 | 89.1 | 85.3 | 77.6 |
| **LidarViT** (ours) | **90.0** | **87.3** | **82.9** | **90.3** | **86.5** | **78.7** |
| MVX-Net-PF(2019) | 86.8 | 86.9 | 81.0 | 87.5 | 84.3 | 74.6 |
| BEVFusion (2022) | 89.5 | 88.9 | 86.3 | 89.0 | 87.1 | 77.4 |
| **FusionViT** (ours) | **91.2** | **90.2** | **88.9** | **90.4** | **88.1** | **79.4** |
| **FusionViT Pretraining** (ours) | **92.1** | **91.4** | **89.9** | **91.2** | **89.5** | **80.8** |

Table 2: Comparison of attained validation mAP (in %) on KITTI with IoU $= 0.7$.

We further use another classical 3D detection dataset KITTI (Geiger et al., 2012) to compare our models' performance with more state-of-the-art methods, which are shown in Table 2. Four cutting-edge frameworks are conducted for pure 3D detection, including the popular point-voxel-based method PV-RCNN (Shi et al., 2021), point-based method Part-$A^2$ (Shi et al., 2020), as well as two latest methods OcTr (Zhou et al., 2023a) and FastPillars (Zhou et al., 2023c). All of these frameworks are outperformed by LidarViT. As for the 3D Fusion track, MVX-Net PointFusion (Sindagi et al., 2019) and BEVFusion (Liu et al., 2022b) are two state-of-the-art Camera-Lidar fusion frameworks. We re-implement them under KITTI settings, and find their performance not as high as our FusionViT's. One possible reason could be due to our promising fusion strategy, which

| Fusion Strategy | Veh. | | Ped. | |
|---|---|---|---|---|
| | AP | APH | AP | APH |
| SUM | 55.5 | 54.3 | 55.1 | 54.6 |
| **CONCAT** | **59.5** | **58.4** | **58.5** | **58.9** |
| DIRECT CONCAT | 57.8 | 57.2 | 56.6 | 56.5 |

Table 3: Ablation Study Results of Different Fusion Strategies On Waymo Open Dataset

| Model | Veh. | | Ped. | |
|---|---|---|---|---|
| | AP | APH | AP | APH |
| Without Both | 37.3 | 36.5 | 38.9 | 37.1 |
| Without LidarViT | 44.1 | 41.4 | 42.7 | 42.9 |
| Without CameraViT | 46.6 | 47.2 | 47.3 | 46.4 |
| Without MixViT | 51.2 | 52.8 | 51.5 | 53.0 |
| **Normal FusionViT** | **59.5** | **58.4** | **58.5** | **58.9** |

Table 4: Ablation Study Results of FusionViT Model Components on Waymo Open Dataset

will be discussed in the next subsection. Additional scores are obtained by the pre-trained version, demonstrating its adaptability. In a work, our FusionViT has a promising performance with excellent resilience under all 2D and 3D detection scenarios in each of the two large-scale datasets.

### 4.3 ABLATION STUDY

We conduct an extensive ablation study and performance analysis next. Firstly we analyze the influence of using different fusion strategies. That's to say, given the learned 2D and 3D represented features, how to combine them more efficiently. To this end, we tried three methods: to SUM, to CONCAT, and DIRECT CONCAT (Cao et al., 2016) the two features. Their performance of them is shown in Table 3. As shown in the second row, use CONCAT operation performs best. That is probably due to that SUM operation is too reckless, ignoring many potentially useful features. DIRECT CONCAT is an efficient method, but using large computation resources and maintained too many features, which may easily cause over-fitting.

We also analyze the influence of the proposed three ViTs in the multi-modal fusion model. We want to show that each sub-model should be important and irreplaceable. To make it, we conduct four more experiments apart from the Normal FusionViT: the FusionViT but without LidarViT component, the FusionViT but without CameraViT component, the FusionViT but without LidarViT and CameraViT components, and the FusionViT but without MixViT components. For the removed components, we add a linear transformation layer to keep the dimension the same in the four experiments. Table 4 shows the results. It is clear the Normal FusionViT has the highest score, indicating that each component of the model is irreplaceable. Particularly, by comparing the fourth and fifth lines, we see the necessity of MixViT. It has about 16% accuracy increase while consuming almost the same training and inference time, compared to directly using a Linear layer to concatenate.

## 5 CONCLUSION

This paper presents FusionViT, a hierarchical Vision Transformer based lidar-camera fusion strategy for 3D object detection. As a pure-ViT based framework, it uses three ViTs to compose its model, so that 2D image features and 3D point cloud features cloud be fused together, learned from each other, and output high-accuracy object detection results. The performance on Waymo Open Dataset and KITTI are promising, which demonstrates that FusionViT is capable for representation learning and object detection tasks studied in this paper.

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

## A EXPERIMENT SETUP

To let the experiment be as close as the real scenarios, we use both Waymo Open Dataset (Sun et al., 2019) and the KITTI (Geiger et al., 2012).

In KITTI, the dataset is divided into 7,481 training samples and 7,518 testing samples. The training samples are commonly divided into a training set (3,712 samples) and a validation set (3,769 samples) following (Chen et al., 2017), which we adopt.

Waymo Open Dataset is the largest open-source dataset in traffic data. It currently contains 798 training sequences and 202 validation sequences. The point clouds are captured with a 64 lanes Lidar, which produces about 180k Lidar points every 0.1s. The official 3D detection evaluation metrics include the standard 3D bounding box mean average precision (mAP) and mAP weighted by heading accuracy (mAPH). The mAP and mAPH are based on an IoU threshold of 0.7 for vehicles and 0.5 for pedestrians.

## B IMPLEMENTATION DETAILS

We implemented DETR, and Swim-Transformer inspired by hungingface (Wolf et al., 2019) library. We used PointPillars, CenterPoint and Deepfusion in lingvo (Shen et al., 2019) packages, and CenterFormer in det3d (Zhu et al., 2019) packages, adopted or re-implemented other comparative models in mmdetection3d (MMDetection3D, 2020) packages. The model is optimized with an Adam (Kingma & Ba, 2014) optimizer with a learning rate of 0.0001 and a dropout rate of 0.3. We train the model on an RTX A6000 GPU for 60 epochs and validate after each epoch.

The voxel grid is defined by a range and voxel size in 3D space. On KITTI, we use $[2, 46.8] \times [-30.08, 30.08] \times [-3, 1]$ for the range and $[0.16, 0.16, 0.16]$ for the voxel size for the $x, y$, and $z$ axes, respectively.

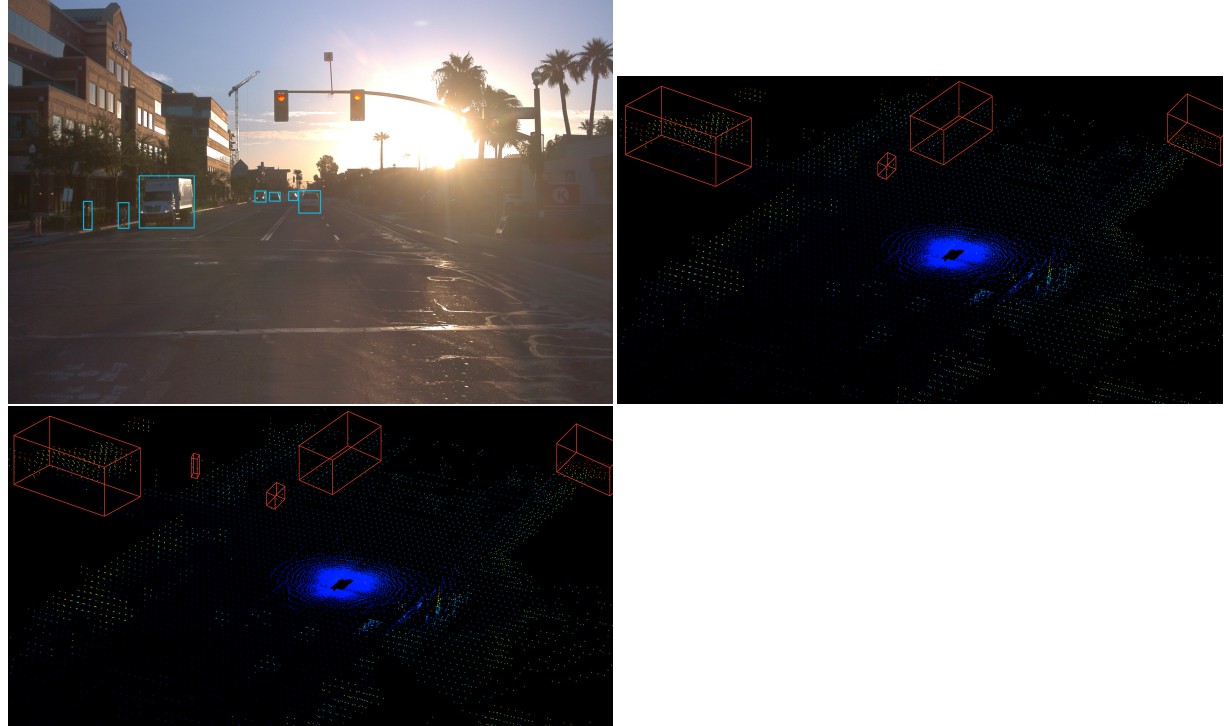

Figure 2: A example showing of results from *upper:* CameraViT; *middle:* LidarViT; and *down:* FusionViT, under same timestamp

Our Waymo datasets uses a detection range of $[-75.2\,\mathrm{m}, 75.2\,\mathrm{m}]$ for the X and Y axis, and $[-2\,\mathrm{m}, 4\,\mathrm{m}]$ for the Z axis. Due to limited computing resources, we randomly sampled 50% of the original training and testing dataset. Only front view perspective from camera and top view perspective from lidar are considered.

Before being fed into the models, both datasets are augmented by random 3D flips, random rotation, scale, and translation, and shuffled the point data.

For the transformer encoder, the $P_c$ is 32, $P_l$ is 64, and $K$ is 6. For the embedding dimension, $D_c$, $D_l$ are both 768, $D_f$ is 1024. As for the input layer dimension, $N_c$, $N_l$ and $N_f$ are all 1024.

In Non-Maximum Suppression (NMS), the threshold is set as 0.3. We let each model can output 256 proposed bounding boxes at most. And for the MLPs, the layers are: [256, 256, 512] in CameraViT, [256, 512, 512] in LidarViT, and [256, 256, 512, 512, 1024] in FusionViT.

## C    CASE STUDY

We conduct visualization by rendering the predicted bounding boxes into point cloud space and images to show the great effectiveness of our fusion model. Figure 2 shows prediction results from pure camera detection, pure lidar detection, and detection using both camera and lidar, under the same timestamp.

Camera images contain lots of textual and color information, which could detect objects in a quite fine manner. Its detection performance is, however, severely dependent on the color and texture diversities, whose performance would be declined significantly in scenes of extreme weather (raining, froggy weather, etc.), mid-night, shadows, reflections, and strong light (sunrise, sunset, etc.), where lots of useful traffic information are devoured by the environment. In the upper plot of Figure 2, lots of small objects could be correctly detected by CameraViT, while the performance decreases dramatically in the part near the strong sunset light. The performance would hardly decrease if using point cloud. It provides detailed information about 3D objects' position, conducting accurate

3D localization, and therefore could maintain detection on a smoothly great level. However, it is difficult to *recognize* an object, leading to relatively low classification performance. In the middle plot of Figure 2, LidarViT could capture the missing objects near the sunset light (like the rightmost one in the picture). It, however, failed to recognize the pedestrian on the left of the street. It could probability detect the existence of the object, but misclassified as an environment (such as the street sign, or trees that have similar size) due to the absence of textual and color information. LidarViT's classification accuracy could also be influenced in other scenes such as the crowded downtown.

With the fusion strategy, the FusionViT takes advantage of strengths in both point cloud and image detection, while complementing each other's weaknesses. As the bottom plot of Figure 2 shows, it could correctly recognize the pedestrian that is missed by LidarViT, using the information provided from the image. It is also robust under different kinds of extreme scenes, with the help of stable 3D space information offered by the point cloud. In addition, by reviewing its prediction performance over a sequence of frames we could find it keeps performing at a high level *under a period*. Unlike single-frame object detection tasks, all inputs are continuously in the driving environment, which means neighborhood frames always have something similar, changing as a gradient-like feature. Originating from the natural language process, the transformer is exactly the model that suits such continuous changes (Vaswani et al., 2017). Therefore, benefiting a lot from spatial and time continuously, our transformer-based detection model could fit well under the traffic scenes.

