# OpenReview forum: "FusionViT: Hierarchical 3D Object Detection via Lidar-Camera Vision Transformer Fusion"
_ICLR.cc/2024/Conference — Submitted to ICLR 2024_

### Official Review · Reviewer_zd5v · 2023-11-01

**Soundness:** 2 fair
**Presentation:** 2 fair
**Contribution:** 1 poor
**Rating:** 3
**Confidence:** 4

**Summary:**

This paper proposes FusionViT, a 3D object detection framework that fuses 2D images with point cloud data. This framework consists of three components, including a 2D image model, a point cloud model, and a fusion model. All three components are based on vision transformer architectures. The method is evaluated on Waymo Open Datset and KITTI benchmarks for 2D & 3D object detections. The results show FusionViT can achieve performance that is competitive with latest works in 2D/3D object detections on those datasets.

**Strengths:**

* This paper is an interesting exploration to use "pure" ViT architectures for 3D object detection. This is a sound research objective as ViT has demonstrated very strong performance in image classification and as a very strong model for visual embeddings. It is generally useful to explore adoption of this backbone in dense prediction in 3D tasks.

* The paper presents a good amount of details and illustrations of the method. For the most part, concepts and algorithms are defined using precise language, assisted with helpful illustrations.

* The proposed method is simple and clean, yet it shows strong performance against baselines that is a reasonable sample of recent works. The benchmarks are done on Waymo dataste and KITTI. Both are popular datasets suitable for evaluation of a 3D object detector.

**Weaknesses:**

* It seems to be an exaggeration to claim that this work is "the first study to investigate the possible pure-ViT based 3D object detection framework".  Transformer based architectures for 3D tasks seem to have been explored extensively in the literature, see [1] for a survey.

* The literature survey in this work is unfortunately not effective. While it lists some of the recent and classical fields, it fails to clearly define the relevance of the current work against the prior approaches. It will help position this work better if such an explicit analysis is presented.

* The paper does not seem to provide much justifications on the various design choices. There are a few well-known 3D object detection paradigms in the literature, voxel based, pillar based and projection based. The paper focuses on the voxel based paradigm. It does not seem to be particularly favorable for a pure ViT architecture. Compared to pillar based approaches it is likely slower and harder to train due to the larger complexity in the attention layers. Compared to a projection based approach it is likely harder to align with the image features (being an 3D detection framework that fuses 2D and 3D). Given this large design space and obvious concerns, I think the paper should provide more rationale and ablation studies to justify the design choices.

* While ViT architectures are shown to be very capable for classification/embedding tasks, it does have a few significant shortcomings in practice. For example, due to lack of the inductive bias it typically requires a large dataset for training to achieve competitive performance. Also, the quadratic complexity in attention layers makes it much harder to be used in dense prediction tasks. These are particularly relevant for 3D object detection using LiDAR as an input modality. It would be reasonable to expect a paper that set out to explore a "pure-ViT based 3D object detection framework" should provide deep analysis on those issues and propose effective mitigations of those shortcomings. It would also be reasonable to expect the same paper to demonstrate the superiority of ViT based architecture versus other sensible architecture choices, such as Swin based methods, despite the possible shortcomings. However, neither is presented in this work.

* The comparison in Table 1 and Table 2 shows promising result of FusionViT compared to some reported numbers of prior works. But as an object detection system, it is critical to provide additional contexts of the accuracy achieved as more expensive systems typically have an edge in terms of accuracy. In this case, I think it is a minimum requirement to compare prior works at similar FLOPs. Better still, latency measured in clearly defined hardware platform should be provided.

[1]Transformers in 3D Point Clouds: A Survey  https://arxiv.org/pdf/2205.07417.pdf

**Questions:**

* Beyond high level number, is there a detailed complexity/accuracy tradeoff provided for the reported results in Table 1 & Table 2?

---

> ### Author Response · Authors · 2023-11-23
> **Thanks for your comments!**
>
> Thank you for your constructive comments and suggestions. For weaknesses and questions mentioned, let us explain them in detail:
>
> 1. As [1] Section 4.1.2 indicates, current transformer-based 3D Object Detection Frameworks are primarily restricted to DETR-based methods, that is, the frameworks with both encoder and decoder (such as 3DETR, CT3D, CAT-Det mentioned in [1]). Designing an encoder-based framework in 3D Object Detection is still a fairly unexplored area.
>
> 2. While to the best of our knowledge it is the first study to investigate the possible pure-ViT-based 3D object detection framework, we accept the suggestion to conduct more exploration on ViT-related issues, such as large datasets for training and the quadratic complexity. We may consider conducting several mathematical analyses and ablation discussions and proposing more strategies to resolve these ViT common issues in future work. We could also report the FLOPs and inference time for a more comprehensive performance comparison.
>
> 3. We clarify that due to space limitations, we cut down several literature surveys of this work. Nevertheless, we tried to outline the connection, pros and cons among the previous related works, as well as their relevance to the current work.
>
> 4. As for comparing with other 3D object detection paradigms, we conducted serval experiments on them. For example, we compared our model with PointPillars and FasterPillars (Pillar-based methods), CenterPoint and Part-{A^2}-free (Point-based methods). For some assumptions such as `Compared to pillar-based approaches it is likely slower and harder to train due to the larger complexity in the attention layers`, and `Compared to a projection-based approach it is likely harder to align with the image features`, my opinion is that we cannot do such general categories' comparison (we could never prove that $a-$based framework must have a good performance (faster, more accurate, etc.) than $b-$based framework, since these framework categories are quite probably. There is no mathematical formula to restrict one method belonging to a group of frameworks. In other words, these frameworks are just categories by Point Cloud input feature preprocessing ways. After all, besides feature preprocessing, it is also important to design an efficient backbone, neck, detection head, and loss function. These efficient models could exist in any kind of data preprocessing category. Therefore, it is not reasonable to do justifications on the various design choices. One could only report methods performance from different such categories.
>
> In light of our clarification and additional results, we would like to ask if the reviewer will reconsider the rating. Thank you!

---

> > ### Comment · Reviewer_zd5v · 2023-12-04
> >
> > I appreciate the response. But there are a few points that I cannot agree with, as I find them to be critical flaws of this paper that must be addressed before it can be accepted.
> >
> > In particular:
> >
> > For point 2, 3D object detection from LiDAR is inherently a practical problem. And one of the most important practical concerns of any object detection systems is the complexity-accuracy tradeoff. A slow but accurate detection system can be much less practical, for example in latency critical applications such as self driving vehicles (where LiDAR based detectors happen to be most commonly applied). I don't think it is a question that can be deferred to a future study.
> >
> > For point 4, it seems the main arguments made by the authors can be summarized as:  theoretical analysis on the pros and cons of an empirical system are usually less useful than thorough empirical comparison in practical metrics, such as accuracy and complexity. The authors have made excellent counter-points to their arguments in point 2 and point 3 in their arguments in point 4. It is also an argument against the conclusion of point 4 itself - if it is difficult to claim a method is superior because it fall into a certain category, than thorough empirical comparison against strong baselines solving the same problems, regardless of the paradigm, is necessary.
> >
> > After looking through the review by other reviewers and the response from the authors, unfortunately I cannot recommend acceptance of this paper.

---

### Official Review · Reviewer_qYip · 2023-11-01

**Soundness:** 2 fair
**Presentation:** 2 fair
**Contribution:** 2 fair
**Rating:** 5
**Confidence:** 5

**Summary:**

The paper proposes a vision transformer based lidar and camera fusion for 3D object detection.
Multi-modal data provides different views of the same scene which makes it more feature-rich compared to single modality models. The paper is motivated by a lack of “pure-ViT” based 3D object detectors and proposes a model that uses independent ViT per modality (CameraViT and LidarViT) to extract single-modality features and fuses them using another ViT (MixViT) and performs bounding box regression and classification on the final output. The CameraViT operates on mini-patches and the LidarViT branch uses voxelization followed by filtering empty voxels and sampling to address the input size problem. The MixViT module operates on the concatenated features to address feature misalignment and modality differences. Experiments are performed on the Waymo Open and KITTI datasets, and show improvement over existing multimodal fusion works.

**Strengths:**

The paper proposes a ViT-only approach for single modality representation learning and  multi-modal fusion in the context of 3D object detection and achieves comparable performance with other camera-lidar fusion based approaches. The lidar-only ViT branch uses voxelization to reduce dimensionality and shows good performance compared to existing lidar-only 3D detectors. The paper also shows ablation studies for the different components which indicates that all the proposed model components are contributing to the performance.

**Weaknesses:**

The paper is motivated by the absence of “pure-ViT” based multi-modal 3D detection models. However, the paper doesn’t explain why such a ViT-only approach is expected to be beneficial for the task.

The overall architecture is not quite novel in that most multi-modal fusion approaches, e.g., DeepFusion, Transfusion, DeepInteraction and any of the BEV fusion based approaches use single modality representation learning followed by multi-modal fusion. It’s unclear what the novelty of the architecture shown in Fig. 1 is.

MixViT uses a large MLP on the concatenated feature which is similar to how DeepFusion uses a localized MLP to learn alignment. The large MLP approach is still learning similar alignment but inefficient in terms of feature utilization.

There’s comparison missing with DeepInteraction (NeurIPS 2022) around the same time as other papers which performs camera-lidar fusion for 3D object detection.

The paper doesn’t show any robustness experiments with lidar-camera spatio-temporal misalignment or robustness of MixViT in the presence of single modality failures.

Performance on NuScenes is also missing in the paper.

Minor
-------
There are a several typos in the paper. For example,
1. "Swim" Transformer written in several places
2. Section 3.5, what is "Multi-Level Perceptions"?

**Questions:**

1. Why is pure-ViT expected to perform better than other approaches in the context of multi-modal 3D object detection?
2. How well does the approach generalize to different data domains. For example, LidarViT to different types of lidar sensors.
3. How does the approach perform under single modality failure and spatio-temporal misalignment?

---

> ### Author Response · Authors · 2023-11-23
> **Thanks for your comments!**
>
> Thank you for your constructive comments and suggestions.  For weaknesses and questions mentioned, let us explain them in detail:
>
> 1. We clarify that, as explained in detail in the Fourth paragraph of Section 1, designing a pure-ViT framework in 3D Object Detection is still a fairly unexplored area. To fill in this gap, we conducted extensive design discussions (Section 3) and experiments (Section 4.3) on it. In the end, we concluded what we found and proposed the presented FusionViT framework.
>
> 2. We clarify that while multi-modal fusion is a general approach that utilizes more than one kind of input for CV tasks (detection), its following structures that embed and process these input features could be much different and require lots of innovation. For instance, DeepFusion emphasizes its two novel techniques: InverseAug and LearnableAlign. Transfusion highlights its soft-association mechanism and image-guided query initialization module. DeepInteraction advocates its modality interaction strategy. Our framework on the other hand promotes our pure-ViT hierarchical architecture.
> We also note that while Transfusion and DeepInteraction use BEV as Lidar input, our method and DeepFusion directly use Point Cloud as the Lidar input.
>
> 3. We agree that the ablation study could be sounder if we add more robustness experiments for MixViT (add `Without LidarViT and MixViT` and `Without LidarViT and MixViT` experiments in Table 4) and for lidar-camera spatio-temporal misalignment. Adding additional comparing experiments for DeepInteraction could also make our paper more convincing. We will add these in the revised version.
>
> 4. KITTI, Waymo Open, and NuScenes are the three most popular datasets for 3D object detection in traffic scenarios. KITTI was first proposed in 2012, it has about 100GB of size which has been well suited for academic research. NuSense and Waymo Open are two industry-level datasets that contain more than 1TB of size for each. They are almost similar in data size, tasks, popularity, and resolution. Due to limited time and devices, we chose to conduct experiments on KITTI and Waymo Open to report the performance under a broader range we could do.
>
> 5. We will correct the typos to have a nicer presentation to readers. Thanks for pointing out!
>
> In light of our clarification and additional results, we would like to ask if the reviewer will reconsider the rating. Thank you!

---

### Official Review · Reviewer_Jv34 · 2023-11-08

**Soundness:** 3 good
**Presentation:** 2 fair
**Contribution:** 3 good
**Rating:** 5
**Confidence:** 4

**Summary:**

This work introduces FusionViT, a transformer architecture that fuses lidar and camera inputs for 3D object detection.
Their model is broadly composed of three components - CameraViT, LidarViT and MixViT.
They train the components in three stages - CameraViT and LidarViT perform 2D and 3D detection, respectively,
and MixViT is trained to fuse the multi-modal features for 3D detection.
They perform experiments on the KITTI and Waymo datasets and show their pure-transformer architecture can outperform other fusion baselines.

**Strengths:**

* LidarViT and lidar transformers are a fairly unexplored area (to my knowledge) and is a solid contribution.
* The high-level idea is straightworward and can be more-or-less summarized by Figure 1.
* Numbers are strong considering the proposed work pivots to a full transformer architecture compared to the baselines.

**Weaknesses:**

Experiment section, besides main results on KITTI and Waymo, not sufficient.
Ablations are not that informative. The paper currently has two ablations now - one which is comparing sum vs. concat for fusion and the other compares removal of LidarViT, CameraViT, MixViT
Model runtime is key for object detection, and with these heavy transformer models, it would good to see some numbers on this.
The authors also introduce a "corner loss" in Section 3.5, which I would have liked to see in the ablations.
What other key design choices were made?

Writing needs improvement. The writing in Sections 3.2, 3.3 is clear enough to understand,
but the mathematical notation is overloaded and actually makes it harder to understand.
An alternative is to summarize Equations 1, 2, 3 with a figure.

**Questions:**

Table 1: from the writing, I assume the first row is performing 2D detection (comparing DETR, Swin, CameraViT).
The second two rows are 3D detection with lidar and lidar-camera fusion, respectively. Why is the first group being compared to the second two?

Is there any difference in the camera/fusion architecture when dealing with single-view (KITTI) and multi-view (Waymo) camera images?
How/is the camera pose information being used in the positional embeddings?

Section 3.6: the authors state they train the model in 3 stages due to "some potential issues of large memory consumption", but still run the model
end to end for training the MixVit? Are the subsequence layers frozen in this stage?

The authors use the word "cubic" to describe the 3D representation of the scene - is there any difference between this term and "voxel"?

Minor typos:
* Swim-transformer
* hungingface

---

> ### Author Response · Authors · 2023-11-23
> **Thanks for your comments!**
>
> Thank you for your constructive comments and suggestions, and they are exceedingly helpful for us to improve our paper. We will improve the writing structure in Section 3 to have a nicer presentation to readers.
>
> For the Experiment section, you mentioned it is not sufficient, could you explain it in more detail? We compared our model with several baselines. For the baseline choosing, we selected both classical and SOTA models, typically for Object Detection tasks in camera-only, lidar-only, and camera-lidar fusion input. In addition, we followed the experiment settings and reported the full evaluation metrics from both KITTI and WOD.
>
> As for Ablations, we chose not to report `Model Runtime`, `The Loss Choices`, and other potential choices since they either do not have a great variation or not being the main research object in this experiment. `Model Runtime` is the first situation. While we will report the general `Model Runtime` in our experiment, it varies nearly negligible under different model structures. `The Loss Choices` is the second situation. The main contribution of this paper is to introduce a novel vision transformer-based 3D object detection model with pure-ViT-based hierarchical architecture. While it could be a future work, we focused on the model structure design, rather than the specific loss function that could match our models best.
>
> For the Questions:
> 1. For the experiments, as mentioned, we selected both classical and SOTA models, typically for Object Detection tasks in camera-only, lidar-only, and camera-lidar fusion input. We put these three groups of experiments together, to show the necessity of using 3D object Detection as well as using fusion. By comparing groups `1 and 2`, we should that 3D Object Detection frameworks generally perform better than pure 2D Detection. From groups `1 and 3` or `2 and 3` we show fusion strategies generally outperform single-modal strategies.
>
> 2. We first clarify that while both KITTI and Waymo have multi-view camera images, we use just a single view for each dataset. For KITTI we used `camera2` and for Waymo we used `front`. Therefore, we do not need to encode the camera pose information.
>
> 3. The subsequence layers of MixVit are not frozen in this stage. For the pertaining, we pre-train CameraViT and LidarViT on the training set. In the fine-tuning, only parameters in CameraViT and LidarViT are frozen.
>
> 4. We clarify that `cubic` is a specification of `voxel`. Firstly introduced from VoxelNet, `voxel` is a general concept that discretizes point clouds. Our LidarViT is a voxel-based framework, which specifically uses `cubic` as the minimal feature representation. The structure and how it is got is explained in detailly in Section 3.3.
>
> In light of our clarification and additional results, we would like to ask if the reviewer will reconsider the rating. Thank you!

---

### Meta-Review · Area_Chair_KCnk · 2023-12-05

**Metareview:**

After the reviews, rebuttal and discussion all reviewers recommend rejection. The main issues raised are evaluation, ablations, and quality of writing. Unfortunately, the authors were not able to sufficiently answer these concerns, and in the end all reviewers were negative. The AC sees no reason to overturn the reviewers recommendataion.

**Justification For Why Not Higher Score:**

No reviewer recommends acceptance.

**Justification For Why Not Lower Score:**

N/A

---

### Decision · Program_Chairs · 2024-01-16

Reject